# Experimental Morogoro Virus Infection in Its Natural Host, *Mastomys natalensis*

**DOI:** 10.3390/v13050851

**Published:** 2021-05-07

**Authors:** Chris Hoffmann, Stephanie Wurr, Elisa Pallasch, Sabrina Bockholt, Toni Rieger, Stephan Günther, Lisa Oestereich

**Affiliations:** 1Bernhard Nocht Institute for Tropical Medicine, 20359 Hamburg, Germany; hoffmann@bnitm.de (C.H.); wurr@bnitm.de (S.W.); pallasch@bnitm.de (E.P.); bockholt@bnitm.de (S.B.); rieger@bnitm.de (T.R.); guenther@bni-hamburg.de (S.G.); 2German Center for Infectious Diseases (DZIF), Partner Site Hamburg, Partner Site Hamburg-Lübeck-Borstel-Riems, Germany

**Keywords:** arenavirus, *Mastomys natalensis*, natural host, virus–host barrier

## Abstract

Natural hosts of most arenaviruses are rodents. The human-pathogenic Lassa virus and several non-pathogenic arenaviruses such as Morogoro virus (MORV) share the same host species, namely *Mastomys natalensis* (*M. natalensis*). In this study, we investigated the history of infection and virus transmission within the natural host population. To this end, we infected *M. natalensis* at different ages with MORV and measured the health status of the animals, virus load in blood and organs, the development of virus-specific antibodies, and the ability of the infected individuals to transmit the virus. To explore the impact of the lack of evolutionary virus–host adaptation, experiments were also conducted with Mobala virus (MOBV), which does not share *M. natalensis* as a natural host. Animals infected with MORV up to two weeks after birth developed persistent infection, seroconverted and were able to transmit the virus horizontally. Animals older than two weeks at the time of infection rapidly cleared the virus. In contrast, MOBV-infected neonates neither developed persistent infection nor were able to transmit the virus. In conclusion, we demonstrate that MORV is able to develop persistent infection in its natural host, but only after inoculation shortly after birth. A related arenavirus that is not evolutionarily adapted to *M. natalensis* is not able to establish persistent infection. Persistently infected animals appear to be important to maintain virus transmission within the host population.

## 1. Introduction

The *Arenaviridae* are a family of single-stranded negative-sense RNA viruses which are predominantly rodent-borne [1]. The majority of this diverse virus family do not cause disease in humans; however, several members can cause severe hemorrhagic fever [2]. One of those pathogenic arenaviruses is Lassa virus (LASV), which poses a serious public health issue in many Western African countries, including Nigeria, Sierra Leone, and Liberia [3], resulting in up to 5000 deaths per year [4].

The natural rodent reservoir of LASV is the natal multimammate rat *Mastomys natalensis*, which often lives in close association with humans and is one of the predominant rodent species in many rural areas of Sub-Saharan Africa [5,6,7]. Similar to other rodent-borne viruses, transmission most likely occurs via contact with rodent excreta or contaminated food, as well as direct contact with infected animals [4]. In nosocomial settings, human-to-human transmission is possible [8], but recent outbreaks seem to be mostly fueled by spill-over from the rodent host [9,10,11].

For a long time, *Mastomys natalensis* (*M. natalensis*) has been assumed to be the sole reservoir for LASV; however, in recent years, several other rodent species have been identified as possible reservoirs [12,13,14]. The wide geographical range of the main reservoir species, as well as the presence of additional host species, heightens the chance of LASV spreading beyond its current domain, putting large populations at risk [3,15]. The lack of specific treatments or licensed vaccines as well as the epidemic potential of this pathogen has led the World Health Organization (WHO) to declare LASV as a priority pathogen in the Research and Development (R&D) Blueprint [16,17]. LASV must be handled in a biosafety level 4 laboratory, further complicating research and the development of countermeasures.

*M. natalensis* is also the rodent reservoir for several other arenaviruses [18,19,20], like Morogoro virus (MORV) in Tanzania. Since MORV is not associated with human disease and shares the same reservoir species, it is an ideal surrogate for modelling LASV infections in the natural rodent host [21]. Mobala virus (MOBV) is another closely related non-pathogenic arenavirus that is endemic in the Central African Republic and has been isolated from *Praomys* sp., which belongs to another rodent genus [22].

Both MORV and LASV seem to be geographically restricted, despite the broad distribution of their rodent reservoirs [15,23]. Additionally, in some locations where LASV was found in other rodent species, *M. natalensis* in the same area does not carry the virus [12,13], further suggesting possible host restrictions.

Several studies have been performed to elucidate the ecology of LASV and MORV in the field [14,23,24,25]; however, only a few experiments have been conducted with the natural rodent host in a laboratory setting [26,27].

Overall, the viral dynamics of LASV and related arenaviruses within the natural hosts remain poorly understood. Gaining deeper insight of the intricate interplay between virus and rodent host is crucial in order to accurately predict future outbreaks and establish successful preventive measures.

In this study, we used MORV as an LASV surrogate to further illuminate the virus–host interactions of African arenaviruses within their natural rodent host. Moreover, we inoculated *M. natalensis* with MOBV as a non-matching virus–host pair to assess possible host restrictions.

## 2. Materials and Methods

### 2.1. Ethics Statement

The study was carried out in strict compliance with the recommendations of the German Society for Laboratory Animal Science under the supervision of a veterinarian. All protocols were approved by the Committee on the Ethics of Animal Experiments of the City of Hamburg (Permit No. 32/14 and N 028/2018). All efforts were made to minimize the number of animals used and to mitigate suffering during experimental procedures. All staff members involved in animal experiments and handling underwent the necessary education and training according to category B or C of the Federation of European Laboratory Animal Science Associations.

### 2.2. Animals and Monitoring

*M. natalensis* were derived from our breeding colony at the Bernhard Nocht Institute for Tropical Medicine (BNITM). All animals descended from breeding pairs that were provided by Heinz Feldmann from the Rocky Mountain Laboratories, Montana, based on an initial colony from wild-caught arenavirus-free animals from Mali. *Mastomys* were housed in small groups in individually ventilated cages. Food and water were accessible *ad libitum*. All animals were monitored regularly for general well-being, signs of disease, and body weight. Humane endpoint criteria included, amongst others, body weight loss of >10% or reduced growth in neonates and juveniles. All experiments were performed with whole litters, and female and male animals were distributed equally to the groups.

The animals were euthanized by isoflurane overdose followed by decapitation, if any of the termination criteria were fulfilled. Additionally, the animals were sacrificed for terminal sampling or at the end of experiments.

A total of 172 animals were used in this study.

### 2.3. Virus Strains

The MORV strain 3017/2004 utilized for this study was isolated at the BNITM [21], whereas the MOBV strain 3099 [22] was obtained from another laboratory. All viruses were grown on Vero E6 cells (ATCC^®^ CRL-1587™, American Type Culture Collection, Manassas, VA, USA) and passaged less than 3 times at the BNITM. Viral stock titers were quantified via immunofocus assay, as described elsewhere [28].

### 2.4. Inoculation

Two-day-old *M. natalensis* neonates were inoculated subcutaneously (s.c.) with 1000 focus forming units (FFU) of either MORV or MOBV in 20 µL PBS. Furthermore, juveniles aged 6, 14, or 27 days were inoculated s.c. with 1000 FFU of MORV in 50 to 100 µL PBS. Infected individuals from these initial infection experiments were used in subsequent transmission experiments.

To assess the impact of natural transmission, breeding pairs were continuously co-housed with previously inoculated offspring. Thus, pregnant females and subsequent litters were in natural contact with virus-shedding individuals. Moreover, naïve four-week-old juveniles were co-housed with chronically infected individuals of the same sex and age.

### 2.5. Sampling and Analysis

Blood, urine, and organs were sampled at frequent intervals up to four months post-infection. Consecutive blood samples were acquired by puncturing the tail vein of juveniles or the saphenous vein for animals older than eight weeks. For blood taken from the saphenous vein, animals were anaesthetized with isoflurane, the fur of the hind leg was shaved, a tourniquet was put on, and the vein was punctured with a safety lancet. Final blood samples were gained via cardiac puncture following euthanasia. Blood was collected in 0.5 mL EDTA tubes (Sarstedt, Germany). Plasma was obtained by spinning the EDTA tubes at 1500 × *g* for 5 min. Urine was collected with Whatman Filter paper (if animals released urine in a restrainer or anesthesia box) or via bladder puncture following euthanasia at terminal sampling points. Urine was extracted from the filter paper by vigorous agitating in 0.2 mL PBS. Heart, spleen, kidney, liver, lung, and brain were taken from all euthanized animals. Gonads were taken from euthanized animals older than 2 weeks. All samples were stored at −20 °C for short-term storage (up to 4 weeks) and −80 °C for long-term storage.

Viral RNA levels in whole blood and urine were analyzed using qRT-PCR assays. RNA extraction was performed using the QIAamp Viral RNA Mini Kit following the manufacturer’s instructions (QIAGEN, Venlo, The Netherlands). The SuperScript™ III Platinum™ One-Step qRT-PCR Kit (Thermo Fisher Scientific, Waltham, MA, USA) was used for the PCR reactions, which were set up based on the protocol described in Nikisins et al. [29]. The details are given in Appendix A. The primer pairs and probes were modified based on the MORV or MOBV L protein sequence (Table 1). All primers and the respective probes were produced by Integrated DNA Technologies, Inc. (Coralville, IA, USA).

In vitro transcripts based on the MORV or MOBV L gene sequence were used to create a standard curve.

Plasma samples were inactivated for serological analysis by mixing them with the same volume PBS containing 2% Triton X-100. The presence of virus-specific IgG antibodies was assessed with indirect immunofluorescence [30]. In short, inactivated sera were diluted 1:50 or 1:100 in PBS and incubated on a monolayer of MORV or MOBV infected Vero E6 cells fixed on slides. Fluorescein-conjugated AffiniPure Goat Anti-Mouse IgG (Jackson Immuno Research, West Grove, PA, USA) was used to visualize virus-specific antibodies in plasma samples.

Sampled organs were homogenized in 1 mL DMEM with 3% FCS using the FastPrep-24^TM^ 5G tissue lysis system with the Lysing matrix D (MP Biomedicals). Infectious virus titers in organs were determined by immunofocus assay as described before [28]. The Old-World arenavirus NP-specific monoclonal antibody 2LD9 [31] was used to detect infected cell foci.

### 2.6. Statistics and Data Presentation

Data presentation and plot preparation was done in Graphpad Prism 9. The body weight gain per day was determined by linear regression. Differences between organ weights were determined by the Mann–Whitney test. The overview figure was created with Biorender.com, accessed on 31 March 2021.

## 3. Results

### 3.1. Infection with MORV

#### 3.1.1. Inoculation of Neonates

To characterize the course of infection after inoculation shortly after birth, one litter of two-day old neonates was inoculated s.c. with 1000 FFU MORV (*n* = 5). The infected neonatal *Mastomys* started to develop viremia at around 1 week post-inoculation (wpi). Antibodies against MORV were detected in plasma from 2 wpi onwards. Furthermore, MORV was also detected in urine and organ samples (Table 2). Despite the presence of antibodies, the animals remained viremic and continuously shed the virus for up to 22 wpi (Figure 1a). Viral titers in blood peaked after 2 wpi. Following a slight decline, they stabilized between 10^6^ and 10^7^ copies/mL and remained at the same level for the duration of the experiments. During the first few weeks, infectious virus was detected in all sampled organs (Figure 2a). The highest viral titers were found in kidneys and lungs during the first wpi (Appendix A). In animals sacrificed 20 wpi and later, viral persistence was limited to few organs, most notably kidney, lung, and testes. No gonads were sampled for animals younger than 2 weeks.

#### 3.1.2. Natural Transmission

To assess the impact of natural transmission, breeding pairs were continuously co-housed with previously inoculated offspring. Individuals from litters born to the same parents that came into contact with their older siblings were also infected with MORV. A total of three litters were exposed via natural contact since birth (*n* = 37) with infected older siblings. These subsequent litters already showed viremia and antibodies shortly after birth (Figure 1b). In some cases, viremia lasted for up to 20 weeks. The highest viral titers were observed in urine samples. However, in contrast to an artificial inoculation, not all individuals showed long-term viral persistence. Only 70% to 80% of individuals from 2 wpi onwards and only 30% of individuals tested after 16 wpi remained PCR-positive in the blood (Table 2). Infectious virus was found in all sampled organs already during the first week of life, most notably in the lung (Figure 2a). Compared to the day 2 s.c. infected animals, the viral titers were lower during the first wpi but increased to a similar level afterwards (Appendix A). No gonads were sampled for animals younger than 2 weeks.

In contrast to the animals that were exposed to infected individuals since birth, four-week old *Mastomys* that were co-housed with chronically age-matched infected individuals (*n* = 9) only developed a transient viremia followed by seroconversion (Table 2).

#### 3.1.3. Age Dependence of Infection

To assess the impact of the host age on the course of infection, one litter each was inoculated s.c. with 1000 FFU MORV at 6 (*n* = 15), 14 (*n* = 10), or 27 (*n* = 10) days of age. *Mastomys* juveniles inoculated at 6 days of age showed viremia 1 wpi, and antibodies were present from 2 wpi onwards (Figure 1c). Animals were followed for four weeks. All tested individuals remained viremic throughout the experiment. However, an overall drop in viral titers and viral clearance from organs for some animals was observed during later sampling points (Figure 2b; Appendix A).

Similarly, MORV inoculation in two-week-old *Mastomys* led to viremia and seroconversion (Figure 1d); however, from 2 wpi onwards, only two-thirds of tested animals remained viremic. No infectious virus was found in organ samples later than 2 wpi (Figure 2b).

Four-week-old *Mastomys* only developed a transient viremia, followed by seroconversion. No infectious virus was detected in organs at any of the sampling time points (Table 2, Appendix A).

Individuals that were inoculated as neonates (day 2) or juveniles (day 6 or day 14) were able to transmit the virus to their parents. However, adult animals only showed transient viremia, followed by seroconversion. An overview with details of all experimental groups is given in Table 2.

### 3.2. Infection with MOBV

To assess possible virus–host restrictions, one litter of two-day-old *Mastomys* neonates was inoculated s.c. with 1000 FFU of the non-matching arenavirus MOBV (*n* = 4). Following the inoculation with MOBV, the animals developed a transient viremia followed by seroconversion. Viral titers in blood were much lower compared to those in MORV-infected animals (Figure 1e). MOBV was also detected in urine and organ samples during the first 2 wpi. After this two-week period, complete viral clearance was observed, and no virus was detected in blood, urine, or organ samples from 4 wpi onwards (Table 2, Appendix A). During their viremic phase, the infected animals were unable to transmit the virus to any of their cage mates (parents and younger siblings). None of the contacts showed viremia or developed virus-specific antibodies (Table 2).

### 3.3. Growth and Development of Infected Animals

Weight, general development (development milestones such as fur growth, opening of the eyes, and development of motor skills), and behavior of the animals were checked every 1–2 days. Inoculated animals were compared to a naive control group (*n* = 54), and 1–4 infected animals per group were sacrificed every week for the determination of organ titers and evaluation of gross pathology (such as malformations, visual bleedings, visual necrosis, enlarged organs). No signs of disease were observed at any time during the experiments. Neither the infection with MORV nor that with MOBV had any adverse effect on the development, growth, and weight gain of the infected individuals (Figure 3a,b). The weight gain of the infected and control animals was on average 0.82 g per day (SD = ±0.02). Animals older than 4 weeks showed increasing differences in weight based on sex, with males being generally heavier than females (Appendix A).

While no differences could be observed for body weight, hearts (Figure 3c,d) and kidneys (Figure 3e,f) of one-week-old *Mastomys* were smaller in the infected animals compared to in the control group. No abnormalities in size or structure of organs or gross pathology were observed otherwise. Already at two weeks, this weight difference was no longer observed, and all groups had similar organ weights.

## 4. Discussion

Similarly, to what has been observed for other arenaviruses such as LCMV, Junin, and Guanarito virus, infection at a young age leads to a long-lasting infection and continuous shedding of the virus in urine [32,33]. Our findings also match observations made by Borremans et al., where *Mastomys* neonates infected with MORV develop a long-lasting viremia in the presence of antibodies. Evidence for MORV persistence in *Mastomys* has also been found in field studies, and mathematic modeling of the data suggests that chronic infections are important for virus maintenance within the host population [34]. We found a clear correlation between the age and the duration of the infection. Only animals that were infected within the first week of their life remained infectious for several weeks, and the percentage of animals that cleared the infection increased with age. Already at an age of four weeks, animals had only transient viremia and cleared the infection within two weeks. Persistently infected individuals, however, were able to horizontally transmit the virus to exposed cage mates, and depending on the age of their cage mates, again, induced a persisting infection (Figure 4). Based on these findings, horizontal or vertical mother-to-offspring transmission seems to be the most likely route of MORV transmission in wild animals, as susceptible individuals (<1 week) would not yet have left the nest, which would reduce their contact with infected animals other than their parents drastically.

Initially, all the tested organs of the infected individuals contained infectious virus with the highest titers usually in kidney and lung. Most persistently infected animals, however, partially cleared the infection, and MORV tended to remain only in kidney, lung, and gonads, although at lower titers. The high rate of MORV-positive kidneys would explain the high virus titers observed in urine and make transmission via urine one of the most likely transmission routes. This is in line with what has been proposed for LASV, where the contamination of food stocks with infectious *Mastomys* urine is discussed as a likely source of human infections [35].

In this study, we could also reproduce the finding that virus-specific IgG antibodies appear approximately 10–14 days post-infection and remain at constant titers for several months. In naturally exposed animals born from seroconverted mothers, antibodies could be detected as early as 1 day post-birth (Figure 2b). The most likely source of these antibodies are maternal antibodies, which appear to be non-protective, as individuals develop viremia despite the antibodies being present before the infection could be established. No correlation between antibody presence and virus clearance could be observed, indicating that the developed antibodies do not play a major role in virus clearance [27]. This finding resembles observation from human Lassa Fever cases, where the presence of antibodies does not appear to be correlated with virus clearance [36]. Neutralizing antibodies, if at all, only appear several months post-infection [37]. Similar observations were made with non-human primates, which develop no or only low levels of neutralizing antibodies after LASV infection [38,39]. Furthermore, experiments with plasma transfer from LASV survivors in non-human primates and humans show variable outcomes, and protection was only achieved in some cases [40,41,42]. These findings further suggest that antibodies are not the driving factor behind viral clearance.

The underlying causes for the observed age dependency of susceptibility and the simultaneous presence of virus and antibodies are the most likely immune tolerance in neonates and juveniles, similarly to what has been described for mice infected with LCMV or murine retrovirus [43,44,45,46]. LCMV-infected neonatal mice also show long-term persistence of virus and have simultaneously virus-specific antibodies. It could be shown that the circulating antibodies in these animals do not exist in a free state but are bound to virus particles or antigens. Studies with LCMV also showed that another key feature of immune tolerance in perinatally infected mice is the unresponsiveness of T cells, especially of virus-specific CD8 T cells [47].

Infection of *M. natalensis* with the matched arenavirus MORV had no lasting impact on the health of the infected animals and all individuals developed normally. This result is expected for a natural virus–host pair and reflects what has been seen for LASV and MORV-infected *Mastomys* in the wild [48]. The weight drop one week post-infection that was observed for MORV-infected *Mastomys* in the study of Borremans et al. could not be reproduced, and one reason could be the different infection route (s.c. versus i.p.) and the different age (2 days to 4 weeks versus 7 to 18 weeks) of the infected animals [27]. In contrast to the infections with host-matched virus MORV, the non-matched virus MOBV was rapidly cleared from the infected neonatal *Mastomys*, indicating an intrinsic host barrier that restricts the replication of arenaviruses with different natural rodent hosts. The presence of infectious MOBV one week after infection and the development of antibodies suggest an active clearance of the virus from the circulation rather than incapability of MOBV to replicate in *Mastomys* cells. Moreover, both MORV and MOBV are capable of infecting interferon-alpha/beta receptor-deficient mice showing a similar disease progression with transient viremia and comparable titers in organs [28], making it less likely that a general attenuation of MOBV is solely accountable for the different infection phenotypes observed in *M. natalensis*. Other potential key players for this clearance could be virus-specific T cells or the innate immune system. T cells have already been shown to be important for LASV, MORV, and MOBV clearance in experimentally infected interferon-alpha/beta receptor-deficient mice [28]. Moreover, they also play a crucial role in the disease progression and outcome of LASV infections in humans and non-human primates [38,39,49,50].

Further studies are needed to explore the underlying cause for the age-dependent susceptibility of *Mastomys* to MORV infections with a special focus on the role of T cells and to elucidate the mechanism of the host barrier restricting the infection of non-matched viruses such as MOBV.

## Figures and Tables

**Figure 1 viruses-13-00851-f001:**
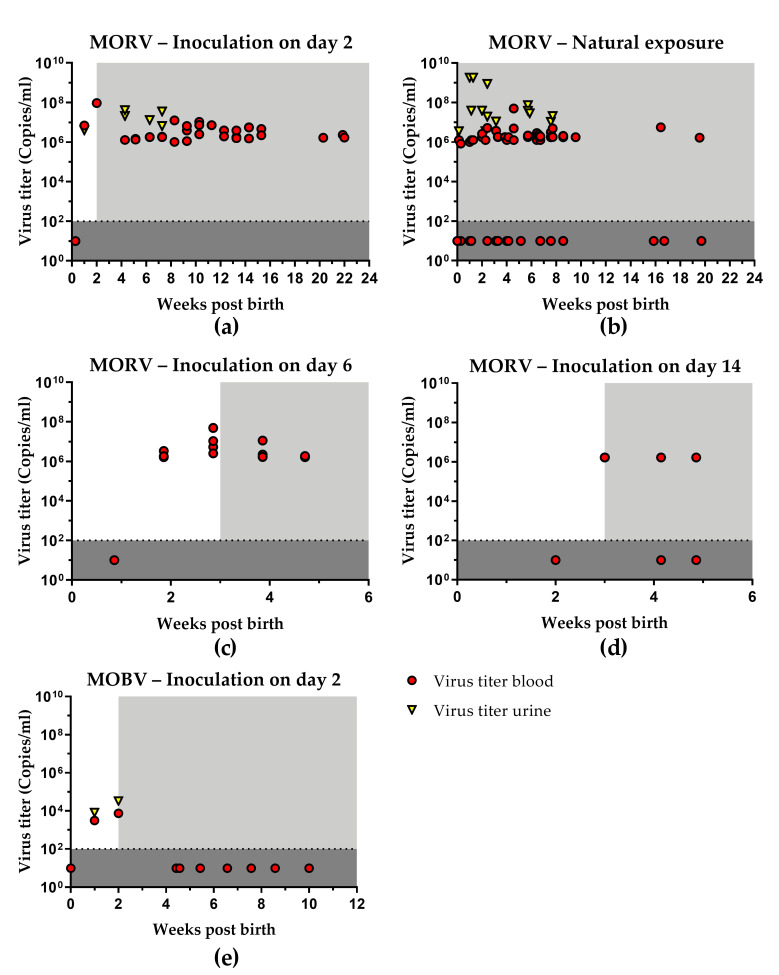
Virus titers in blood and urine of infected *Mastomys*. Neonates were inoculated with MORV at the age of 2 days (**a**) or exposed to the virus (**b**) via direct contact with infected individuals. Furthermore, *Mastomys* were inoculated with MORV at 6 days (**c**) or 14 days (**d**) post-birth. (**e**) Neonates were inoculated with MOBV at the age of 2 days. Blood and urine were collected at regular intervals and tested for the presence of MORV or MOBV RNA with qRT-PCR. Ct values were converted into copy numbers using a standard curve. Blood samples are depicted as red dots and urine samples were indicated by yellow triangles. Plasma was inactivated and analyzed for the presence of MORV- or MOBV-specific antibodies with indirect immune fluorescence. The presence of virus-specific antibodies is marked by the light grey area. The limit of detection for the qRT-PCR assay is shown by the dotted line and dark grey coloration. Negative samples have been assigned a default value below the detection limit.

**Figure 2 viruses-13-00851-f002:**
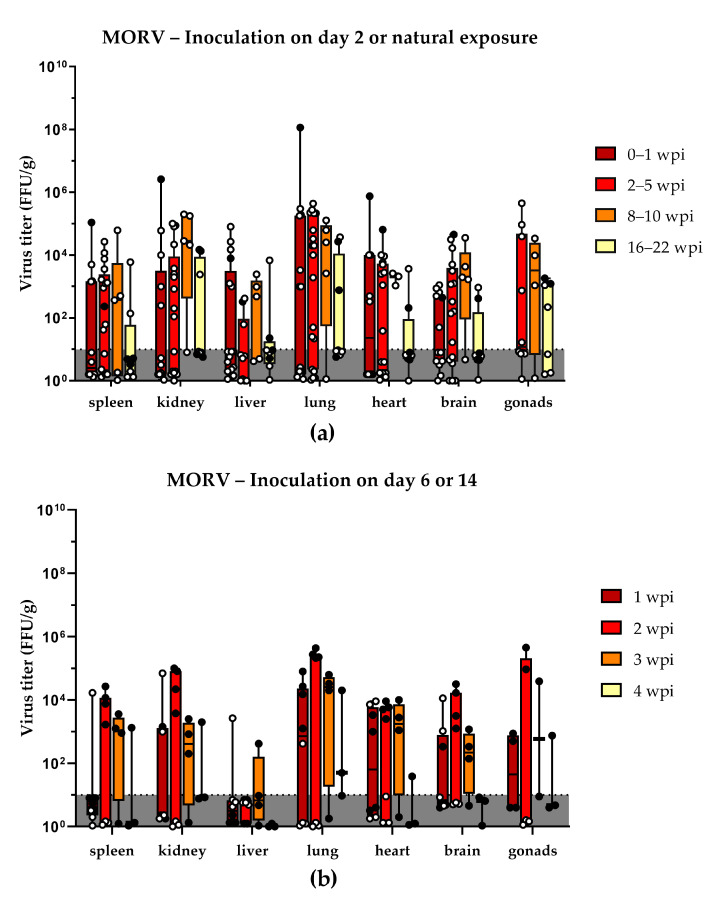
Organ titers of MORV-infected animals: (**a**) virus titers of organs from *Mastomys* that have been infected via inoculation 2 days post-birth (black dots) or via natural contact with infected individuals (white dots); (**b**) virus titers of organs from animals that have been inoculated 6 days (black dots) or 14 days (white dots) post-birth. The different time points post-infection are shown in different colors, and all samples falling into this time frame have been pooled. The limit of detection is shown by the dotted line and dark grey coloration. Titers are shown as box and whisker plots with all data points shown. Virus titers of negative samples (below the limit of detection) appear to vary due to the differences in organ weight, which is taken into account for the virus titer per gram organ.

**Figure 3 viruses-13-00851-f003:**
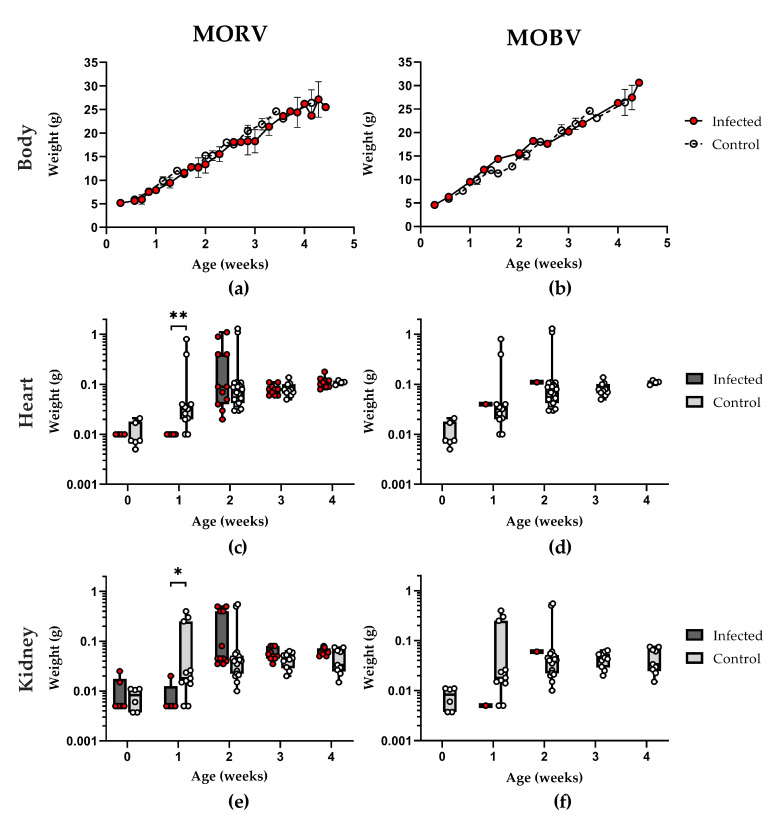
Growth and development of infected *Mastomys*. Data for the MORV-infected animals (inoculated subcutaneously (s.c.) with 1000 focus forming units (FFU) on days 2, 6, and 14 as well as naturally exposed) are shown on the left, and data for the MOBV-infected animals (inoculated s.c. with 1000 FFU on day 2) are shown on the right. (**a**,**b**) Body weight, as well as organ weight for hearts (**c**,**d**) and kidneys (**e**,**f**) have been measured during the first four (MORV) or two (MOBV) weeks of life. The infected individuals are depicted by red dots, whereas the uninfected control group is depicted by white dots. The body weight is depicted as mean with SD, and organ weights are shown as box and whiskers with all data points plotted. Statistical differences in weight are indicated by * *p* < 0.05 and ** *p* < 0.01, as determined by the Mann–Whitney test.

**Figure 4 viruses-13-00851-f004:**
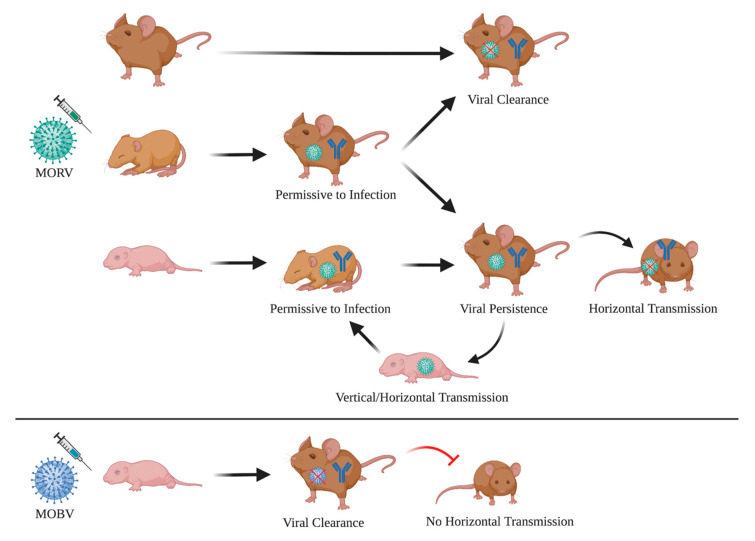
Schematic summary of the findings. Susceptibility of *Mastomys* to MORV infections is age-dependent. Neonates are permissive to the infection, can become chronically infected and transmit the virus horizontally, leading to more chronically infected animals. Adults and weaned individuals are permissive to the infection with MORV but clear the virus within a few weeks. Infected juveniles show a mixed phenotype with some individuals remaining long-term infected and some clearing the virus, although at a slower rate compared to adults. All animals that are in contact with the virus develop IgG antibodies, independent of their infection status. MOBV-inoculated neonatal *Mastomys*, on the other hand, show only transient viremia, clear the virus rapidly and are unable to transmit the virus. The figure was created by BioRender.com.

**Table 1 viruses-13-00851-t001:** PCR primer and probe sequences.

Primer/Probe ^1^	Sequence (5′ → 3′)
Nikisins F2 MORV	AAT CAA TTT GTG AAT GTG CCA
Nikisins R MORV	GCT CAG GTT TCA TAT AGT TTA GAC CA
Nikisins TM MORV	/56-FAM/AAG TGG GGC/ZEN/CCA ATG ATG TCC CCA TT/3′ IB^®^FQ/
Nikisins F2 MOBV	AAC CAA CTT ATG GAT ATG CCA
Nikisins R MOBV	TGG GCC TTC TAT CTT ATA GCC TGG ACC A
Nikisins TM MOBV	/56-FAM/AAT GGG GGC/ZEN/CTA TGA TGA CCC CCT T/3′ IB^®^FQ/

^1^ Probes = 250 nm PrimeTime^®^ 5′ 6-FAM™/ZEN™/3′ IB^®^FQ.

**Table 2 viruses-13-00851-t002:** Overview of the infection status of *Mastomys* following inoculation or exposure with Morogoro virus (MORV) or Mobala virus (MOBV).

		Blood		Urine	Organs
Group	Sampling Period (Weeks Post-Birth)	PCR Positive Samples/Tested Samples ^1^ per Sampling Period	Ab Positive Samples/Tested Samples ^1^ per Sampling Period	PCR Positive Samples/Tested Samples ^1^ per Sampling Period	Virus Positive Animals/Tested Animals ^2^ per Sampling Period
MORV inoculation day 2 (*n* = 5)	1	1/1	0/1	1/1	1/1
	2	1/1	1/1	n.t.	1/1
	4–6	9/9	9/9	3/3	n.t.
	7–9	8/8	7/8	1/1	n.t.
	10–12	8/8	8/8	n.t.	n.t.
	13–15	8/8	8/8	n.t.	n.t.
	20–22	3/3	3/3	n.t.	3/3
MORV inoculation day 6 (*n* = 15)	2	4/4	0/4	n.t.	4/4
	3	4/4	4/4	n.t.	4/4
	4	4/4	4/4	n.t.	3/4
	5	3/3	3/3	n.t.	2/3
MORV inoculation day 14 (*n* = 10)	3	4/4	1/4	n.t.	4/4
	4	2/3	3/3	n.t.	0/3
	5	2/3	3/3	n.t.	n.t.
MORV inoculation day 27 (*n* = 10)	5	0/2	1/2	n.t.	0/2
	6	0/2	1/2	n.t.	n.t.
	7	0/2	2/2	n.t.	n.t.
	8	0/2	2/2	n.t.	0/2
	9	1/2	2/2	n.t.	n.t.
MORV exposure to persistently infected inviduals from birth (*n* = 37)	0–1	6/10	8/10	4/9	5/12
	2–3	10/14	14/14	6/8	6/8
	4–7	26/32	31/32	2/4	0/4
	8–10	14/20	20/20	2/3	4/5
	16–20	2/6	6/6	n.t.	2/5
MORV exposure to persistently infected inviduals from day 25 (*n* = 9)	4–5	7/9	2/9	n.t.	n.t.
	6–7	5/18	18/18	0/3	n.t.
	8	0/9	9/9	0/7	0/3
MORV exposure of adults to infected individuals (*n* = 14)	0–1	0/10	3/10	n.t.	n.t.
	2–4	3/15	9/15	n.t.	0/1
	5–9	1/8	8/8	n.t.	0/2
	11–13	0/2	2/2	n.t.	0/2
MOBV inoculation day 2 (*n* = 4)	1	1/1	0/1	1/1	0/1
	2	1/1	1/1	1/1	1/1
	4–7	0/6	6/6	0/4	n.t.
	8–10	0/6	6/6	0/2	0/2
MOBV exposure to infected individuals from birth (*n* = 12)	1–4	0/12	0/8	n.t.	0/4
	5–7	0/8	0/8	n.t.	n.t.
MOBV exposure of adults to infected individuals (*n* = 2)	1–2	0/3	0/3	n.t.	n.t.
	5–10	0/3	0/3	n.t.	n.t.

^1^ For blood and urine, the number of positive samples versus the total number of samples tested during a given time period is shown. Since individuals were sampled up to 3 times per sampling period, the number of tested samples can exceed the number of animals per group. ^2^ For organs, the number of positive animals versus the total number of tested animals is shown. Animals were considered positive, if one or more of the tested organs contained infectious virus. n.t. = not tested.

## Data Availability

The data presented in this study are available on request from the corresponding author.

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
