# Peer review of "Experimental Morogoro Virus Infection in Its Natural Host, Mastomys natalensis"

_viruses, 2021, doi:10.3390/v13050851_

Round 1
Reviewer 1 Report
This is an interesting paper with a lot of very useful information about the infection dynamics of arenaviruses in their natural hosts. These types of studies are rare, yet vital for understanding the epidemiology of these viruses. I think the study has been conducted with great care and rigor and is scientifically sound.
I have a few minor comments.
Please clarify the number of animals in the different categories that were scarified at the various time points and whose organ titer results are presented; it is not clear now to me what the sample size is per time point.
I suggest mentioning somewhere that also Borremans et al., 2015 found that neonates inoculated with MORV establish a persistent infection in the presence of antibodies. I also suggest discussing the findings in Mariën et al., 2020, as they found evidence for chronically infected Mastomys natalensis in nature.
Marien, J., B. Borremans, C. Verhaeren, L. Kirkpatrick, S. Gryseels, J. Gouy de Bellocq, S. Gunther, C. A. Sabuni, A. W. Massawe, J. Reijniers, and H. Leirs. 2020. 'Density dependence and persistence of Morogoro arenavirus transmission in a fluctuating population of its reservoir host', Journal of Animal Ecology, 89: 506-18.
Line 56. The way it is written here it makes it seem as if the particular species from which MOBV was isolated has been identified; while identification had only been possible at the genus level (Praomys).
Table 2. How is it possible that for ‘MORV inoculation day 2’ the number tested (often 8) is higher than the number of individuals inoculated? If it this is due to multiple samples tested for some individuals, please clarify.
Line 177. ‘Individuals from litters born to the same parents that came into contact with their older siblings were also infected with MORV.’: this sentence gives the impression the older siblings were also inoculated with MORV?
Line 210. Please show the data behind the statement “adult animals only showed transient viremia, followed by seroconversion”. Why not add it to the table?
Line 260. Please show the data behind the statement “Animals older than 4 weeks showed increasing differences in weight based on gender with males being generally heavier than females”.
Author Response
Dear Mr. Everett,
Please find enclosed our reply to the comment from Reviewer 1. In our response, the main points of a reviewer’s critique are presented in regular type, followed by our response in italic. All changes in the manuscript are highlighted in yellow and one additional supplementary figure S1 has been added.
Please clarify the number of animals in the different categories that were scarified at the various time points and whose organ titer results are presented; it is not clear now to me what the sample size is per time point.
The information about the number of animals per group can be found in the tables 2, S2 and S3. The number of animals that were sacrificed at the different time points can be found in the tables S2 and S3. Organs were analyzed from all sacrificed animals. We recognize that the number of blood samples, analyzed for PCR and serology was confusing as animals were sampled up to three times per sampling period. We changed the header of the table 2 to “PCR Positive samples/tested samples1 per sampling period” and added the following footnote: “For blood and urine, the number of positive samples versus the total number of samples tested during a given time period is shown. Since individuals were sampled up to 3 times per sampling period the number of tested samples can exceed the number of animals per group.”
We hope this change makes it easier for the reader to understand who many animals we had in the different groups.
I suggest mentioning somewhere that also Borremans et al., 2015 found that neonates inoculated with MORV establish a persistent infection in the presence of antibodies.
Thank you very much for this comment, we added the following sentence to the manuscript: “Our findings also match observations made by Borremans et al. where Mastomys neonates infected with MORV developed a long lasting viremia in the presence of antibodies.
I also suggest discussing the findings in Mariën et al., 2020, as they found evidence for chronically infected Mastomys natalensis in nature.
Thank you very much for this comment, we added the following sentence and the Mariën et al., 2020 publication as a reference to the manuscript: “Our findings also match observations made by Borremans et al. where Mastomys neonates infected with MORV developed a long-lasting viremia in the presence of antibodies. Evidence for MORV persistence in Mastomys has also been found in field studies and mathematic modeling of the data suggests that chronic infections are important for virus maintenance within the host population.”
Line 56. The way it is written here it makes it seem as if the particular species from which MOBV was isolated has been identified; while identification had only been possible at the genus level (Praomys).
Thank you very much for this comment, we changed the sentence from “another rodent species” to “which belongs to another rodent genus” in the manuscript.
Table 2. How is it possible that for ‘MORV inoculation day 2’ the number tested (often 8) is higher than the number of individuals inoculated? If it this is due to multiple samples tested for some individuals, please clarify.
Thank you very much for this comment. We recognize that the tables showing the infection status we confusing. We changed the header of the table to better reflect, that the data is given as positive samples out of all samples taken and that individual were samples up to three times in any given sampling period. We also added a table footnote to make clear that individuals were tested up to 3 times in any given sampling period.
Line 177. ‘Individuals from litters born to the same parents that came into contact with their older siblings were also infected with MORV.’: this sentence gives the impression the older siblings were also inoculated with MORV?
We changed the sentence from “co-housed with infected offspring” to “co-housed with previously inoculated offspring” to make it clearer, that for this section, we analyzed animals that were naturally in contact with inoculated (= experimentally infected) cage mates. The corresponding sentence in the materials and methods section has also been changed accordingly. “To assess the impact of natural transmission, breeding pairs were continuously co-housed with previously inoculated offspring. Thus, pregnant females and subsequent litters were in natural contact with virus-shedding individuals.”
Line 210. Please show the data behind the statement “adult animals only showed transient viremia, followed by seroconversion”. Why not add it to the table?
Thank you very much, we added the groups to the table 2.
Line 260. Please show the data behind the statement “Animals older than 4 weeks showed increasing differences in weight based on gender with males being generally heavier than females”.
We added a figure showing the weight curves for male and female to the supplement and changed the reference in the manuscript accordingly.
In summary, we have responded to all comments and have modified the manuscript accordingly. We thank the reviewers for the constructive criticism and hope that the revised version is now fully acceptable for publication in Viruses.
Best regards,
Lisa Oestereich

Reviewer 2 Report
This manuscript, by Hoffmann et al., showed experimentally the Morogoro virus infection in its natural host, Mastomys natalensis at different age and measured the health status of the animal, virus load in blood and organs, the development of virus-specific antibodies, and the ability of infected individuals to transmit the virus. The authors provided data that showed animals infected with Morogoro virus as neonate up to 2 weeks after birth developed persistent infection albeit specific antibody production and were continuously shedding the virus and were able to transmit the virus horizontally. In contrast, infected adults and weaned individuals did not develop chronic infection and cleared the virus within a few weeks. In addition, infections between not-matched MOBV/Mastomys natalensis neonates also induced virus specific antibody production, but unlike the MORV infection, the MOBV was rapidly cleared by the neonates and were not able to transmit to others.
The manuscript was well written, and the data logically presented.
The major problem with this manuscript is the total lack of attempt for a plausible mechanistic insight, unlike the authors claimed in their abstract “we investigated the history of infection and mechanisms of virus transmission……”. Other than mentioned in the discussion that the underlying causes for the observed age dependency for persistent infection is most likely due to immune tolerance in neonates, there is neither attempt experimentally nor theoretically to address it. Since both MORV and MOBV infection produce specific antibodies, but MORV induces persistent infection in neonate while MOBV doesn’t, what are the known structural differences between MORV and MOBV? Do these structural differences known to induce/suppress immune tolerance? Relevant online search and proper discussion should be added. On the other hand, are CD4 and CD8 T cell numbers different in the neonates after MORV and MOBV infection? Could the T cell number differences account for the immune tolerance? In addition, the authors showed that there was no correlation between antibody presence and virus clearance, that the developed antibodies don’t play a major role in virus clearance, why? Is it due to antibody to virus ratio? If the MORV infected neonates were provided with additional antibodies, could they clear the infection and become non-infectious?
It is important for the authors to answer these questions to strengthen this manuscript for the publication.
Author Response
Dear Mr. Everett,
Please find enclosed our reply to the comment from Reviewer 2. In our response, the main points of a reviewer’s critique are presented in regular type, followed by our response in italic. All changes in the manuscript are highlighted in yellow and one additional supplementary figure S1 has been added.
The major problem with this manuscript is the total lack of attempt for a plausible mechanistic insight, unlike the authors claimed in their abstract “we investigated the history of infection and mechanisms of virus transmission……”.
Thank you very much for this comment. We removed the “mechanism” from the abstract as we indeed did not study the mechanism of transmission. The sentence line 12 now reads: “In this study, we investigated the history of infection and virus transmission within the natural host population.”
Other than mentioned in the discussion that the underlying causes for the observed age dependency for persistent infection is most likely due to immune tolerance in neonates, there is neither attempt experimentally nor theoretically to address it. Since both MORV and MOBV infection produce specific antibodies, but MORV induces persistent infection in neonate while MOBV doesn’t, what are the known structural differences between MORV and MOBV? Do these structural differences known to induce/suppress immune tolerance? Relevant online search and proper discussion should be added.
Unfortunately, no detailed study has been published comparing MOBV with other closely related arenaviruses. Both viruses have an identical genome organization but only about 70% sequence identity. As neither MORV nor MOBV are characterized on a structural level, it is extremely difficult to speculate about the role of individual mutations on replication or immune evasion. Furthermore, no in vivo or in vitro study has been published that analyses differences in e.g. the ability of MORV/MOBV to induce/suppress innate or adaptive T cell responses. Publications looking at differences in immune activation/suppression usually compare the pathogenic Lassa virus with the apathogenic Mopeia virus. There is only one publication showing that MOBV and MORV are both able to infect Interferon-alpha/beta receptor-deficient mice and both viruses show similar disease progression. Both viruses showed transient viremia and comparable although low titers in organs. Upon T cell depletion, both viruses were able to establish chronic infection but no major differences could be observed between the two viruses. We expanded the discussion and added published data to show that the observed effects between MORV and MOBV are most likely not due to a general attenuation of MOBV.
“Moreover, both MORV and MOBV are capable of infecting interferon-alpha/beta receptor-deficient mice showing a similar disease progression with transient viremia and comparable titers in organs [28], making it less likely that a general attenuation of MOBV is accountable for the different infection phenotypes observed in M. natalensis.”
On the other hand, are CD4 and CD8 T cell numbers different in the neonates after MORV and MOBV infection? Could the T cell number differences account for the immune tolerance?
We tested more than 20 monoclonal and broadly reactive antibodies that recognize rat, human, and mice T cell surface antigens. Unfortunately, no commercially available anti CD3/4/8 antibody that we screened was cross-reactive with M. natalensis T cells, preventing the in-depth analysis of the T-cell response. Efforts are currently being made to produce M. natalensis specific monoclonal antibodies to analyze the T cell response in the future.
We added a paragraph to the discussion highlighting that T cells are most likely driving forces behind the virus clearance and that further research in this area is needed. “T cells have already been shown to be important for LASV, MORV, and MOBV clearance in experimentally infected interferon-alpha/beta receptor-deficient mice [28]. Moreover, they also play a crucial role in the disease progression and outcome of LASV infections in humans and non-human primates [38, 39, 49, 50].”
In addition, the authors showed that there was no correlation between antibody presence and virus clearance, that the developed antibodies don’t play a major role in virus clearance, why? Is it due to antibody to virus ratio? If the MORV infected neonates were provided with additional antibodies, could they clear the infection and become non-infectious?
It is well known, that LASV survivors (human and non-human primates) develop no or if at all very late (several months post disease) neutralizing antibodies. Most antibodies found in different studies appear to be raised against the nucleoprotein and play no role in protection. Plasma transfer experiments in non-human primates showed very variable outcome and protection was only achieved in some cases. This points towards antibodies generally not playing a major role for long-term protection. The study of Marien et al. 2020 could also show that antibodies and virus can co-exist in the same animal in the wild. To give a better comparison to the data known from humans and NHPs we expanded the following paragraph to the discussion:
“No correlation between antibody presence and virus clearance could be observed, indicating that the developed antibodies do not play a major role in virus clearance [27]. This finding resembles observation from human Lassa Fever cases, where presence of anti-bodies does not appear to be correlated with virus clearance [36]. Neutralizing antibodies, if at all, only appear several months post-infection [37]. Similar observations were made with non-human primates, which develop no or only low levels of neutralizing antibodies after LASV infection [38, 39]. Furthermore, experiments with plasma transfer from LASV survivors in non-human primates and humans show variable outcomes, and protection was only achieved in some cases [40-42]. These findings further suggest that antibodies are not the driving factor behind viral clearance.”
In summary, we have responded to all comments and have modified the manuscript accordingly. We thank the reviewers for the constructive criticism and hope that the revised version is now fully acceptable for publication in Viruses.
Best regards,
Lisa Oestereich

Round 2
Reviewer 2 Report
The authors have addressed all of my concerns.